# Evaluating Scald Reactions of Some Turkish Barley (*Hordeum vulgare* L.) Varieties Using GGE Biplot Analysis

Kadir Akan [1,*], Ahmet Cat [2] , Onur Hocaoglu [3] and Mehmet Tekin [4]

1 Department of Plant Protection, Faculty of Agriculture, Kırşehir Ahi Evran University, 40100 Kırşehir, Türkiye
2 Department of Plant Protection, Faculty of Agriculture, Siirt University, 56100 Siirt, Türkiye; ahmetcat@siirt.edu.tr
3 Department of Field Crops, Faculty of Agriculture, Çanakkale 18 Mart University, 17020 Çanakkale, Türkiye; onurhocaoglu@comu.edu.tr
4 Department of Field Crops, Faculty of Agriculture, Akdeniz University, 07059 Antalya, Türkiye; mehmettekin@akdeniz.edu.tr
* Correspondence: kadirakan@ahievran.edu.tr

**Abstract:** Scald caused by the fungal pathogen *Rhynchosporium commune* is a significant foliar disease affecting barley production on a global scale, and it leads to substantial reductions in both yield and quality of barley. In the current study, the reactions of 40 Turkish barley (*Hordeum vulgare* L.) varieties to scald were evaluated under natural conditions in Çanakkale and Kırşehir in 2021–2022, and Antalya and Siirt locations in 2022–2023 growing seasons. Field trials were conducted according to randomized block design with three replications in each year; the spore concentration was $1 \times 10^6$ spores per mL, and it was applied to the varieties three times at different growth stages. The reactions of barley varieties were assessed using a newly designed two-digit scale ranging from 11 to 99. Based on their scale values, the varieties were categorized as immune (0), resistant (11–35), moderately resistant (36–55), moderately susceptible (56–75), and susceptible (76–99). In addition, genotype plus genotype-by-environment (GGE) interactions of scale values were analyzed through GGE Biplot and explained 97.65% of the total variation. The ranking of genotypes based on scale groups generally showed consistency with GGE Biplot results, but GGE Biplot offered a more detailed classification, especially for moderately susceptible varieties. The relationship between the two methods indicated the relative stability of variety reactions, as GGE Biplot analysis also considered genotype stability. In conclusion, the use of the newly developed scale for evaluating scald reactions in barley gives reliable results. In addition, identified resistant varieties can serve as valuable genetic resources for further breeding studies.

**Keywords:** barley; scald; *Rhynchosporium commune*; disease reaction; GGE Biplot

## 1. Introduction

Scald, caused by the fungus *Rhynchosporium commune* (formerly known as *Rhynchosporium secalis*), is a highly destructive disease affecting not only barley but also other *Hordeum* species and *Bromus diandrus* [1]. This widespread disease has been recorded in more than 50 countries and can cause a significant reduction in grain yield under favorable conditions, with losses of up to 40% in susceptible varieties. It is characterized as a fungal disease that can be transmitted through seed and stubble and develops under wet and cool growing conditions [2]. Growing susceptible barley varieties in a debris-retaining harvesting system also contributes to its favorability. The fungus can infect the host plant at various stages of growth, resulting in visible symptoms on leaves, leaf sheaths, and ears of barley [3]. Economically, it is important as a foliar disease, affecting regions in central and western Asia, North Africa, Europe, the Americas, and Australia. Its presence results in significant yield losses and a decline in grain quality [4–6]. On average, yield losses can range from 1% to 19%, although reported losses have been as high as 10% to 70% [7,8]. In Türkiye, several

studies have reported losses ranging from 1% to 31% [9], and between 8.9% and 30.5% [10] due to this disease. Among the various components of yield, the number of ears per square meter was the most adversely affected [11].

Kavak [10] reported that the disease significantly reduced the 1000-grain weight in some barley varieties. In addition to these, in a study conducted by Çelik and Karakaya [12], a survey was conducted in the Eskişehir province of Türkiye during the 2012 growing season. They examined 121 barley fields and determined that the disease was present in 108 of these fields. The average prevalence of the disease across these fields was determined to be 22.7%. Similarly, Özdemir et al. [13] investigated 128 fields in Kırıkkale province of Türkiye in 2017, 117 of which were infected with scald. The findings from this survey indicated that the prevalence of the disease in the province was 4.37%.

The control of scald in susceptible barley varieties often relies on the application of fungicides [14–16]. However, it is important to note that many farmers may face financial constraints due to the high cost of using fungicide. Therefore, alternative strategies are essential to mitigate losses and enhance economic returns for producers. One of the most economical approaches to managing this disease is the development and usage of barley varieties that are resistant or tolerant [17]. This strategy not only reduces the reliance on costly fungicides but also contributes to sustainable and environmentally friendly farming practices. To achieve this, it is crucial to conduct screening of germplasm for resistance to scald. Related with this, Düşünceli et al. [18] screened 683 barley genotypes at adult plant stage for resistance to scald and reported that 39% of the genotypes gave resistance or a moderate resistance reaction. Mert and Karakaya [19] evaluated scald reactions of 37 Turkish barley varieties and 2 candidate varieties at the seedling stage. The results of the study indicated that seven varieties and one candidate variety were resistant.

Several disease assessment scales have been used to determine the level of the host reaction to the pathogen. For instance, in their research, McLean and Hollaway [20] used a 1–9 scale, Çelik and Karakaya [12] assessed disease severity using the 1–9 scale developed by Saari and Prescott [21], and Xue et al. [22] used a 0–9 scale for disease evaluation at adult-plant stage, where 0 represents no disease and 9 represents severe leaf damage. According to Xue et al. [22], scores <2.1 indicate resistance reaction, scores between 2.1 and 3.0 indicate moderate resistance reaction, and scores >3.1 indicated susceptible reaction. In another study, Kavak [10] used a 0–100 scale to evaluate variety reactions, classifying them as very resistant (0–5), resistant (5–10), susceptible (10–50), and very susceptible (50–100). Additionally, Düşünceli et al. [18] assessed variety reaction using five groups: 0–3.0 as resistant (R), 3.1–4.0 as moderately resistant (MR), 4.1–6.0 as moderately susceptible (MS), 6.1–8.0 as susceptible (S), and 8.1–9.0 as highly susceptible (HS).

The objectives of this study were as follows: (i) to determine the reactions of some Turkish barley varieties against scald, (ii) to identify the disease reaction groups such as immune, resistant, moderately resistant, moderately susceptible, and susceptible, and (iii) to evaluate the identified reaction groups by performing GGE Biplot analysis.

## 2. Materials and Methods

### 2.1. Plant Materials and Field Trials

A collection of 40 Turkish barley varieties were used as plant materials in this study (Table 1). This collection, including 33 winter and 7 spring varieties, was evaluated for resistance to scald, 31 of which are the two-row type and the remainder of which are the six-row type.

The field trials were conducted with randomized block design with three replications at Kırşehir and Çanakkale locations during the 2021–2022 growing season, and Antalya and Siirt locations during the 2022–2023 growing season. The genetic materials were sown in two-meter rows for each replication. To evaluate the reaction of each variety, a local susceptible variety "Aydanhanım" was sown every 10 rows. In addition, cultural practices were performed manually during field trials.

**Table 1.** Information about Turkish barley varieties used in this study.

| No | Variety | Row Type | Registration Holder | Registration Year |
|---|---|---|---|---|
| 1 | Tokak 157/37 | Two-rowed | Central Research Institute for Field Crops | 1963 |
| 2 | Zafer 160 | Six-rowed | Central Research Institute for Field Crops | 1964 |
| 3 | Yeşilköy 387 | Six-rowed | Trakya Agricultural Research Institute | 1967 |
| 4 | Cumhuriyet 50 | Two-rowed | Transitional Zone Agricultural Research Institute | 1973 |
| 5 | Yerçil-147 | Two-rowed | Transitional Zone Agricultural Research Institute | 1976 |
| 6 | Quantum | Two-rowed | Faculty of Agriculture, Ege University | 1983 |
| 7 | Obruk 86 | Two-rowed | Central Research Institute for Field Crops | 1986 |
| 8 | Anadolu 86 | Two-rowed | Central Research Institute for Field Crops | 1986 |
| 9 | Bülbül 89 | Two-rowed | Central Research Institute for Field Crops | 1989 |
| 10 | Erginel 90 | Six-rowed | Transitional Zone Agricultural Research Institute | 1990 |
| 11 | Şahin-91 | Two-rowed | GAP International Agricultural Research Institute | 1991 |
| 12 | Tarm-92 | Two-rowed | Central Research Institute for Field Crops | 1992 |
| 13 | Bornova 92 | Six-rowed | Aegean Agricultural Research Institute | 1992 |
| 14 | Efes-3 | Two-rowed | Anadolu Efes | 1992 |
| 15 | Yesevi 93 | Two-rowed | Central Research Institute for Field Crops | 1993 |
| 16 | Orza 96 | Two-rowed | Central Research Institute for Field Crops | 1996 |
| 17 | Balkan 96 (Igri) | Two-rowed | Trakya Agricultural Research Institute | 1996 |
| 18 | Karatay 94 | Two-rowed | Bahri Dağdaş International Agricultural Research Institute | 1996 |
| 19 | Kalaycı-97 | Two-rowed | Transitional Zone Agricultural Research Institute | 1997 |
| 20 | Kıral-97 | Six-rowed | Bahri Dağdaş International Agricultural Research Institute | 1997 |
| 21 | Beyşehir | Two-rowed | Bahri Dağdaş International Agricultural Research Institute | 1998 |
| 22 | Konevi | Two-rowed | Bahri Dağdaş International Agricultural Research Institute | 1998 |
| 23 | Sladoran | Two-rowed | Trakya Agricultural Research Institute | 1998 |
| 24 | Şerifehanım 98 | Two-rowed | Aegean Agricultural Research Institute | 1998 |
| 25 | Vamıkhoca 98 | Six-rowed | Aegean Agricultural Research Institute | 1998 |
| 26 | Akhisar 98 | Six-rowed | Aegean Agricultural Research Institute | 1998 |
| 27 | Anadolu 98 | Two-rowed | Anadolu Efes | 1998 |
| 28 | Efes 98 | Two-rowed | Anadolu Efes | 1998 |
| 29 | Angora | Two-rowed | Anadolu Efes | 1999 |
| 30 | Çetin 2000 | Six-rowed | Central Research Institute for Field Crops | 2000 |
| 31 | Çumra 2001 | Two-rowed | Anadolu Efes | 2001 |
| 32 | Çatalhüyük 2001 | Two-rowed | Anadolu Efes | 2001 |
| 33 | Akar | Two-rowed | Central Research Institute for Field Crops | 2012 |
| 34 | Avcı-2002 | Six-rowed | Central Research Institute for Field Crops | 2002 |
| 35 | Çıldır 02 | Two-rowed | Transitional Zone Agricultural Research Institute | 2002 |
| 36 | Sur-93 | Two-rowed | GAP International Agricultural Research Institute | 2002 |
| 37 | Zeynel Ağa | Two-rowed | Central Research Institute for Field Crops | 2003 |
| 38 | Başgül | Two-rowed | Anadolu Efes | 2003 |
| 39 | İnce-04 | Two-rowed | Transitional Zone Agricultural Research Institute | 2004 |
| 40 | Fahrettinbey | Two-rowed | Black Sea Agricultural Research Institute | 2004 |

### 2.2. Collection of Isolates and Single Spore Isolation

Barley leaves infected with scald were collected in barley growing areas from Antalya, Çanakkale, Kırşehir, and Siirt provinces of Türkiye. These leaves were firstly dried in a paper envelope at 24 °C for one week. Dried leaves were sliced into 2 mm sections and soaked in sterile distilled water for 5 min, then subjected to surface sterilization using 70% of ethanol for 15 s, and finally treated with a 0.5% sodium hypochlorite solution for 90 s. After this process, the leaf segments were rinsed with sterile distilled water twice and dried into sterile filter paper for 1 min [23]. These dried leaves were placed on Petri dishes containing Bean Agar (BA) medium (140 g of green beans, 20 g dextrose, 18 g agar, and 1 L distilled water) supplemented with streptomycin (50 mg per liter) and storage at 24 °C in an incubator. After three weeks, the fungal colony was observed, and each colony was transferred to new medium to obtain the single-spore isolates. After obtaining single-spore cultures, they were stored at 4 °C until use.

### 2.3. Inoculation, Incubation, and Disease Assessment

To obtain the inoculum, each isolate was grown in BA medium for two weeks. Later, distilled water was added onto the colony, and spores were collected. This spore concentration was cleaned from other parts of the colonies using sterile cheesecloth and the final volume was prepared as $1 \times 10^6$ spore per mL. Then, 1 mL of Tween-20® was added to each 100 mL of inoculum.

The barley varieties were inoculated three times from the beginning to the end of the tillering stage with two-weeks intervals [23]. Disease evaluation was made at the milk development stage [24]. In the disease evaluation, the highest score among the replications was recorded for each variety and the scoring was performed using modified Saari and Prescott's double-digit scale (00–99) [21], representing the severity of scald. In this scale, the first digit, denoted as D1, provides the relative height of the disease symptoms on the plant and corresponds to the vertical disease progression using the original Saari-Prescott scale, ranging from 0 to 9. The second digit, referred to as D2, pertains to the severity of the disease and is measured in terms of the infected leaf area. In this study, a scale modified from Saari and Prescott [21] ranging from 11 to 99 was used. To facilitate classification and analysis, the two-digit values were re-adjusted as 0, 11–35, 36–55, 56–75, and 76–99. They were considered as immune, resistant, moderately resistant, moderately susceptible, and susceptible, respectively.

*2.4. Statistical Analysis*

Analysis of variance (ANOVA) is used to test the significance of genotype environment interactions. Reaction scores were subjected to arcsine transformation in order to stabilize variance for ANOVA [25]. Scald reactions of barley varieties were evaluated with genotype plus genotype-by-environment (GGE) Biplot analysis. GGE Biplot is a well-established statistical method to evaluate genotype environment interactions [26]. It is a multivariate method which approximates multi-environmental data into a single data matrix using singular value decomposition which produces unique eigenvalues for the data, then selects the best two of these eigenvalues in terms of their ability to explain the variation within the data and projects them on a biplot [27]. The output graphics can be considered as 2D summary including genotypes and environments on the same plane and enables the comparison of environments, selecting the best performing and most stable genotypes in all environments combined or selecting the best-performing genotypes for any given environment [28]. In this study, we analyzed the scald reactions of 40 varieties in four different environments: E1: Çanakkale, E2: Kırşehir, E3: Antalya, and E4: Siirt. The statistical model of the GGE Biplot was as follows:

$$Y_{ij} - \mu - \beta_j = \lambda_1\, \xi_{i1}\, \eta_{1j} + \lambda_2\, \xi_{i2}\, \eta_{2j} + \varepsilon_{ij} \qquad (1)$$

where $Y_{ij}$ = expected scald value of genotype *i* in environment *j*, $\mu$ = mean of all genotype environment combinations, $\beta_j$ = main effect of environment *j*, $\lambda_1$ and $\lambda_1$ are the singular values of the first and second largest principal components (PC1 and PC2), $\xi_{i1}$ and $\xi_{i1}$ are eigenvectors of genotype *i* when $\eta_{1j}$ and $\eta_{2j}$ are the eigenvectors of environment *j* for PC1 and PC2, respectively, and lastly, $\varepsilon_{ij}$ = the residue for each genotype environment combination that were not explained by PC1 and PC2.

GGE Biplot analysis was conducted using the 'GGEBiplots' package v 0.1.3 in R environment (3.6.2) [29], which implements the original methodology [27]. Genotype rankings were evaluated on an GGE Biplot with row (genotype) preserving singular value decomposition (SVD), as recommended [30,31]. Genotype—Environment relationships were evaluated on a different GGE Biplot with symmetrical SVD which scaled by standard deviation. Apart from the different SVD preferences, both GGE Biplots were identical in terms of data input (same data without any transformation) and centering (tester, G + GE).

**3. Results**

Scald reactions of barley varieties were observed when the susceptible variety (Aydanhanım) reached the infection value of 85 at least. The high infection value indicates that the reaction tests in the study were successfully performed. According to the two years with four different locations, the pathogen infection was observed in different ratios at adult plant stage and these results are given in Table 2. Based on the scald reactions of barley varieties in the 2021–2022 and 2022–2023 growing seasons, the varieties Yeşilköy 387, Çetin

2000, Zafer 160, Avcı-2002, Kıral-97, and Erginel 90 showed resistance reactions at all four locations. (Table 2).

**Table 2.** Scald reactions of 40 Turkish barley varieties to *R. commune* at adult-plant stage.

| No | Varieties | Row Type | E1 | E2 | E3 | E4 |
|----|-----------|----------|----|----|----|----|
| 1 | Tokak 157/37 | Two-rowed | 87 | 94 | 93 | 44 |
| 2 | Zafer 160 | Six-rowed | 21 | 11 | 12 | 19 |
| 3 | Yeşilköy 387 | Six-rowed | 11 | 18 | 10 | 15 |
| 4 | Cumhuriyet 50 | Two-rowed | 88 | 84 | 85 | 92 |
| 5 | Yerçil-147 | Two-rowed | 75 | 74 | 69 | 73 |
| 6 | Quantum | Two-rowed | 82 | 83 | 83 | 84 |
| 7 | Obruk 86 | Two-rowed | 57 | 93 | 84 | 82 |
| 8 | Anadolu 86 | Two-rowed | 83 | 75 | 91 | 91 |
| 9 | Bülbül 89 | Two-rowed | 92 | 83 | 91 | 83 |
| 10 | Erginel 90 | Six-rowed | 21 | 11 | 10 | 30 |
| 11 | Şahin-91 | Two-rowed | 86 | 79 | 82 | 72 |
| 12 | Tarm-92 | Two-rowed | 84 | 83 | 93 | 81 |
| 13 | Bornova 92 | Six-rowed | 84 | 73 | 84 | 77 |
| 14 | Efes-3 | Two-rowed | 94 | 73 | 93 | 82 |
| 15 | Yesevi 93 | Two-rowed | 92 | 83 | 92 | 83 |
| 16 | Orza 96 | Two-rowed | 94 | 83 | 85 | 74 |
| 17 | Balkan 96 (Igri) | Two-rowed | 71 | 64 | 65 | 62 |
| 18 | Karatay 94 | Two-rowed | 94 | 76 | 84 | 81 |
| 19 | Kalaycı-97 | Two-rowed | 92 | 82 | 93 | 83 |
| 20 | Kıral-97 | Six-rowed | 10 | 23 | 19 | 12 |
| 21 | Beyşehir | Two-rowed | 81 | 86 | 73 | 92 |
| 22 | Konevi | Two-rowed | 93 | 84 | 93 | 81 |
| 23 | Sladoran | Two-rowed | 74 | 61 | 72 | 55 |
| 24 | Şerifehanım 98 | Two-rowed | 84 | 75 | 92 | 82 |
| 25 | Vamıkhoca 98 | Six-rowed | 82 | 62 | 84 | 63 |
| 26 | Akhisar 98 | Six-rowed | 31 | 20 | 21 | 49 |
| 27 | Anadolu 98 | Two-rowed | 94 | 85 | 93 | 94 |
| 28 | Efes 98 | Two-rowed | 94 | 82 | 95 | 85 |
| 29 | Angora | Two-rowed | 84 | 71 | 83 | 73 |
| 30 | Çetin 2000 | Six-rowed | 11 | 29 | 10 | 8 |
| 31 | Çumra 2001 | Two-rowed | 93 | 83 | 92 | 82 |
| 32 | Çatalhüyük 2001 | Two-rowed | 94 | 83 | 93 | 75 |
| 33 | Akar | Two-rowed | 93 | 86 | 92 | 88 |
| 34 | Avcı-2002 | Six-rowed | 9 | 12 | 19 | 23 |
| 35 | Çıldır 02 | Two-rowed | 76 | 65 | 75 | 63 |
| 36 | Sur-93 | Two-rowed | 85 | 74 | 84 | 81 |
| 37 | Zeynel Ağa | Two-rowed | 93 | 82 | 92 | 83 |
| 38 | Başgül | Two-rowed | 94 | 82 | 85 | 84 |
| 39 | İnce-04 | Two-rowed | 93 | 84 | 92 | 85 |
| 40 | Fahrettinbey | Two-rowed | 74 | 63 | 75 | 64 |

E: Environments; E1: Çanakkale, E2: Kırşehir, E3: Antalya, and E4: Siirt.

ANOVA results indicated the existence of significant genotype x environment interactions (Table 3). The GGE Biplot analysis, using the Singular Value Decomposition (SVD) method with symmetric scaling (standard deviation), explained 97.65% of the total variation to assess genotype-environment interaction (Figure 1).

**Table 3.** Combined ANOVA results of scald reactions of the varieties over four locations.

| Source | DF * | Sum of Squares | F Ratio |
|---|---|---|---|
| Genotype | 39 | 110,871.6 | 140.855 ** |
| Environment | 3 | 2026.39 | 33.4671 ** |
| Replication [Environment] | 8 | 1550.48 | 9.6027 ** |
| G × E | 117 | 19,281.36 | 8.1652 ** |
| Error | 312 | 6297.06 | |
| Total | 479 | 140,026.89 | |
| C.V. (%) | 7.87 | | |
| $R^2$ | 0.96 | | |

* DF: Degree of freedom, ** Significant at $p < 0.01$ level.

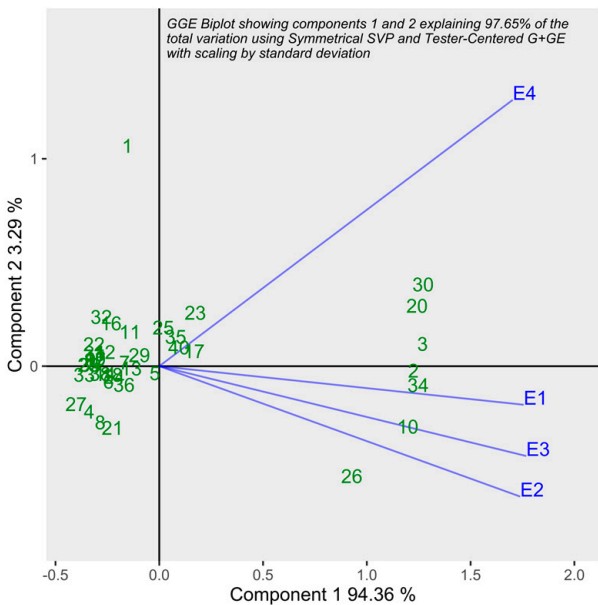

**Figure 1.** Symmetrical GGE Biplot of scald reactions of barley varieties. (The green colored numbers in the figure are the variety numbers. The variety numbers are as given in Table 1).

Proximity of genotypes to the environment indicated that pre-selected barley varieties could be recommended for the related location. This result can be further confirmed in the genotype ranking presented in Table 4. The variety Yeşilköy 387 ranked first in mean scald reactions (MSR) when Çetin 2000 ranked second in all environments. Varieties Zafer 160 and Avcı-2002 both ranked third in MSR followed by Kıral-97 (ranked 4th) and Erginel 90 (ranked 5th). In terms of stability across environments, Yeşilköy 387 and Zafer 160 ranked higher (5th and 11th, respectively) by the lowest standard deviations (SD) among all varieties. SD rankings of Avcı-2002, Kıral-97 and Erginel 90 varied between 17 and 31 when susceptible varieties such as Quantum, Yerçil-147, Akar, and Cumhuriyet 50 were ranked as first, second, third, and fourth, respectively. The reactions of Turkish barley varieties to scald were assessed using a GGE Biplot with a focus on genotypes, as given in Figure 2.

This Biplot accounted for 97.65% of the total variance, with the majority of it being attributed to PC1. In this graph, a hypothetical "average environment axis" (AEA) passing through the origin of the Biplot represented an average environment. Varieties positioned in close proximity to the AEA were considered stable. When the arrow falls within the innermost circle, it indicates the variety with the highest scald resistance and stability. According to this criterion, GGE Biplot analysis revealed the existence of 10 variety groups. The previously marked three varieties (Yeşilköy 387, Zafer 160, Avcı-2002) were clustered together within the innermost circle and were categorized into the first group (Figure 2). This classification was a result of their remarkable scald reaction and stability across the environments.

The second group had Kıral-97 and Erginel 90 with Çetin 2000 variety being the sole member of the third group. Similarly, the fourth group included only variety Akhisar 98. First four groups of the GGE Biplot had resistant varieties (Scale Group 2) when GGE Biplot groups 5 and 6 did not include any varieties. Similarly, moderately susceptible (Scale group 4) and susceptible (Scale Group 5) varieties were dispersed to groups from GGE Biplot group 7 to 10. Overall, the GGE Biplot facilitated a more comprehensive assessment by dividing the resistant varieties into the first four groups when also separating moderately susceptible and susceptible varieties into four groups (Table 4). The distribution of genotypes in the groups of the GGE Biplot (Figure 2) was in relation with the scale groups, especially when it came to identifying the best varieties, as shown in Table 4. This consistency suggests the stability of scald-resistant varieties since GGE Biplot analysis simultaneously selects for high performance and stability.

**Table 4.** Rankings of Turkish barley varieties based on their reactions to *R. commune*.

| No | Varieties | MSR * | MSR Rank | SD | SD Rank | Scale Group | GGE Group |
|----|-----------|-------|----------|-----|---------|-------------|-----------|
| 3 | Yeşilköy 387 | 13.5 | 1 | 3.7 | 5 | 2 | 1 |
| 30 | Çetin 2000 | 14.5 | 2 | 9.75 | 31 | 2 | 3 |
| 2 | Zafer 160 | 15.75 | 3 | 4.99 | 11 | 2 | 1 |
| 34 | Avcı-2002 | 15.75 | 3 | 6.4 | 20 | 2 | 1 |
| 20 | Kıral-97 | 16 | 4 | 6.06 | 17 | 2 | 2 |
| 10 | Erginel 90 | 18 | 5 | 9.42 | 30 | 2 | 2 |
| 26 | Akhisar 98 | 30.25 | 6 | 13.45 | 34 | 2 | 4 |
| 17 | Balkan 96 (Igri) | 65.5 | 7 | 3.87 | 6 | 4 | 7 |
| 23 | Sladoran | 65.5 | 7 | 9.04 | 29 | 4 | 7 |
| 40 | Fahrettinbey | 69 | 8 | 6.38 | 19 | 4 | 7 |
| 35 | Çıldır 02 | 69.75 | 9 | 6.7 | 22 | 4 | 8 |
| 5 | Yerçil-147 | 72.75 | 10 | 2.63 | 2 | 4 | 8 |
| 25 | Vamıkhoca 98 | 72.75 | 10 | 11.87 | 33 | 4 | 8 |
| 29 | Angora | 77.75 | 10 | 6.7 | 22 | 5 | 9 |
| 7 | Obruk 86 | 79 | 11 | 15.43 | 35 | 5 | 9 |
| 13 | Bornova 92 | 79.5 | 12 | 5.45 | 14 | 5 | 9 |
| 1 | Tokak 157/37 | 79.5 | 12 | 23.87 | 36 | 5 | 10 |
| 11 | Şahin-91 | 79.75 | 13 | 5.91 | 16 | 5 | 9 |
| 36 | Sur-93 | 81 | 14 | 4.97 | 10 | 5 | 9 |
| 6 | Quantum | 83 | 15 | 0.82 | 1 | 5 | 9 |
| 21 | Beyşehir | 83 | 15 | 8.04 | 26 | 5 | 10 |
| 24 | Şerifehanım 98 | 83.25 | 16 | 6.99 | 23 | 5 | 9 |
| 18 | Karatay 94 | 83.75 | 17 | 7.59 | 24 | 5 | 9 |
| 16 | Orza 96 | 84 | 18 | 8.21 | 27 | 5 | 9 |
| 8 | Anadolu 86 | 85 | 19 | 7.66 | 25 | 5 | 10 |
| 12 | Tarm-92 | 85.25 | 20 | 5.32 | 13 | 5 | 10 |
| 14 | Efes-3 | 85.5 | 21 | 9.95 | 32 | 5 | 9 |
| 38 | Başgül | 86.25 | 22 | 5.32 | 13 | 5 | 10 |
| 32 | Çatalhüyük 2001 | 86.25 | 22 | 9 | 28 | 5 | 10 |
| 4 | Cumhuriyet 50 | 87.25 | 23 | 3.59 | 4 | 5 | 10 |
| 9 | Bülbül 89 | 87.25 | 23 | 4.92 | 9 | 5 | 10 |
| 15 | Yesevi 93 | 87.5 | 24 | 5.2 | 12 | 5 | 10 |
| 19 | Kalaycı-97 | 87.5 | 24 | 5.8 | 15 | 5 | 10 |
| 31 | Çumra 2001 | 87.5 | 24 | 5.8 | 15 | 5 | 10 |
| 37 | Zeynel Ağa | 87.5 | 24 | 5.8 | 15 | 5 | 10 |
| 22 | Konevi | 87.75 | 25 | 6.18 | 18 | 5 | 10 |
| 39 | İnce-04 | 88.5 | 26 | 4.65 | 8 | 5 | 10 |
| 28 | Efes 98 | 89 | 27 | 6.48 | 21 | 5 | 10 |
| 33 | Akar | 89.75 | 28 | 3.3 | 3 | 5 | 10 |
| 27 | Anadolu 98 | 91.5 | 29 | 4.36 | 7 | 5 | 10 |

* MSR: Mean scald reaction, SD: Standard deviation. Scale groups 2: Resistant, 4: Moderately Susceptible, 5: Susceptible. Varieties were arranged and scale groups were assessed on MSR values of each variety.

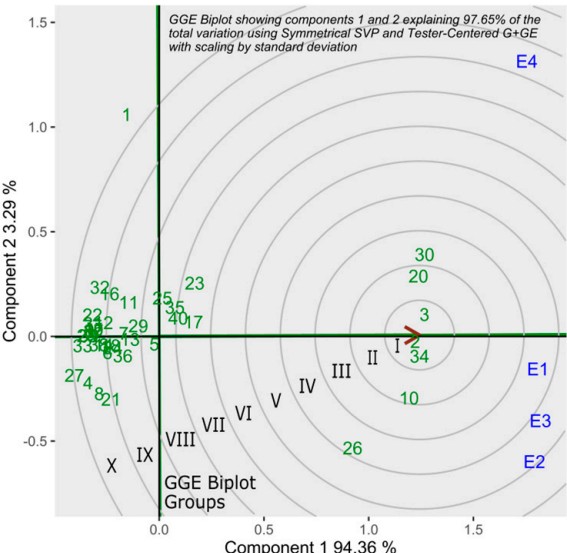

**Figure 2.** GGE Biplot of scald reactions of barley varieties for ranking genotypes (The green colored numbers in the figure are the variety numbers. The variety numbers are as given in Table 1).

## 4. Discussion

Barley scald, caused by the fungal pathogen *Rhynchosporium commune*, is indeed a significant disease that affects barley crops globally. Scald can lead to reduced crop yields and quality, making it a major concern for barley farmers and the agricultural industry. Therefore, the importance of finding new sources for resistance to this pathogen is needed. Developing resistant/tolerant barley varieties is a crucial strategy for managing and mitigating the impact of the disease. Furthermore, the scale used for assessing scald at the adult-plant stage are inadequate, with various scales being performed by several researchers [12,18,22]. In this study, we assessed two- and six-row-type barley varieties for resistance to scald at adult-plant stage, as well as the suitability of the recently developed scale with disease assessments being evaluated using GGE Biplot analysis and categorized from immune to susceptible reactions.

Based on disease scoring, it has been observed that six-row Turkish barley varieties had lower reaction values to *R. commune* compared to those of two-row varieties. This finding is correlated with the results obtained in various studies [11,18,19,32,33]. In addition, Albustan et al. [34] determined that only the variety Erginel 90 was determined resistant to *R. commune* among the 15 barley varieties tested in reaction studies conducted in both field and greenhouse conditions. Another study conducted by Mert and Karakaya [19] reported that there was a high variation in the reactions of barley varieties and differences in the pathogenicity of five *R. commune* isolates. In the present study, among the varieties subjected to reaction tests, Yeşilköy 387, Zafer 160, Avcı-2002, Kıral-97, Erginel 90, Çetin 2000, and Akhisar 98 were determined resistant in all test locations. Similarly, Düşünceli et al. [18] identified that the barley varieties Avci 2002, Çetin 2000, Kıral 97, Erginel 90, Akhisar 98, Kaya 7794, Yeşilköy 387, and Zafer 160 were resistant. On the other hand, the Vamikhoca, Çıldır 02, and Quantum varieties were determined as susceptible in greenhouse tests but they had resistant reactions at adult-plant stage. In total, 25 of them were also determined susceptible to scald in both greenhouse and field conditions. In our study, we obtained comparable outcomes to those reported in other studies [33], demonstrating differentiation in reactions to scald among the varieties. In this study, 27 of 40 varieties showed susceptible reactions when Vamikhoca and Çıldır 02 were moderately resistant. These varieties had been previously documented in various studies by several researchers [19,23,33], and the findings are in accordance with the current research in general.

Different statistical methods have been used in different plant species to determine genotype-environment interaction and to identify superior genotypes exhibiting broad or

specific adaptation to different environments through multi-location trials. Recently, the GGE biplot method has been increasing popularity over other methods due to its better explanation of genotype-environment interaction and its easy-to-understand approach. This analysis has been extensively preferred through multi-location trials to screen powdery mildew, leaf rust, spot blotch, and fusarium head blight diseases [35–39] to identify stable and durable resistant plant materials. However, it has not yet been used for the evaluation of genotypes against scald in barley. In this study, this method was used to identify the resistance varieties and compare the ranking groups among the varieties according to the new scale. Several studies showing the genotype and environment interaction in relation to plant disease reactions have also observed comparatively greater stability in resistance genotypes [40–42]. These results are consistent with the findings of this study.

## 5. Conclusions

Some Turkish barley varieties are used in resistance breeding programs as resistance sources especially for their scald resistance. In this study, seven barley varieties (Yeşilköy 387, Zafer 160, Avcı-2002, Kıral-97, Erginel 90, Çetin 2000, and Akhisar 98) were determined as resistant to scald. This result shows the important potential of varieties for resistance to this disease. On the other hand, it is remarkably important that all varieties with resistant reactions were six-rowed when two-rowed barley varieties had moderately susceptible or susceptible reactions in both years. Assessment of genotype stability is a requirement for the field trials conducted in multiple years and locations. In this study, the results of GGE Biplot reveals the insights of the multi-environmental data as a whole, complementing the scale assessment which does not relate to genotype environment interaction. Thus, our results suggest that using the GGE Biplot to determine scald reactions of Turkish barley varieties is a practical approach and provide convenient characterization of barley varieties grown under varying environments.

**Author Contributions:** Conceptualization, K.A. and A.C.; methodology, K.A. and A.C.; software, K.A., A.C., O.H. and M.T.; validation, K.A., A.C. and O.H.; formal analysis, K.A., A.C. and O.H.; investigation, K.A., A.C., O.H. and M.T.; resources, K.A., A.C., O.H. and M.T.; data curation, K.A., A.C., O.H. and M.T.; writing—original draft preparation, K.A., A.C. and O.H.; writing—review and editing, A.C., M.T., K.A. and O.H.; visualization, K.A. and O.H.; All authors have read and agreed to the published version of the manuscript.

**Funding:** This research received no external funding.

**Data Availability Statement:** The data presented in this study are available on request from the corresponding author.

**Acknowledgments:** The authors thank to the Scientific Research Projects Coordination Unit of Kırşehir Ahi Evran University for continuous support.

**Conflicts of Interest:** The authors declare no conflict of interest.

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
