# Peer review of "Evaluating Scald Reactions of Some Turkish Barley (Hordeum vulgare L.) Varieties Using GGE Biplot Analysis"

_agronomy, doi:10.3390/agronomy13122975_

Round 1
Reviewer 1 Report
Comments and Suggestions for Authors
In this manuscript, authors used GGE Biplot to evaluate the scald reaction of 40 Turkish Barley varieties under four locations.
There are some comments that required a response from the authors.
[1] Is there a significant difference in their scald reactions (the level of host response to the pathogen) between 2021-2022 and 2022-2023 growing seasons? Give a statistical result.
[2] Please offer a detail of the four test sites, e.g., latitude, longitude, elevation (m), and annual rainfall (mm).
[3] ANOVA and multiple comparison should be used to assess the scald reaction of these Turkish Barley varieties.
[4] The full name of GGE should be “genotype plus genotype-by-environment” (L134-135).
[5] The subscripted index should be in the equation (1), e.g., ɛij.
[6] The version of GGEBiplots should be added (L155).
Author Response
Thank you very much for taking the time to review this manuscript. Please find the detailed responses below and the corresponding revisions/corrections in red changes in the revised file.
In this manuscript, authors used GGE Biplot to evaluate the scald reaction of 40 Turkish Barley varieties under four locations.
There are some comments that required a response from the authors.
[1] Is there a significant difference in their scald reactions (the level of host response to the pathogen) between 2021-2022 and 2022-2023 growing seasons? Give a statistical result.
We added an ANOVA indicating significant difference between growing seasons and genotype x environment interactions. (Line 176)
[2] Please offer a detail of the four test sites, e.g., latitude, longitude, elevation (m), and annual rainfall (mm).
As the disease was artificially inoculated and its development was induced by moisture pressure in trials, the influence of climate can be disregarded. Therefore, we do not think that is necessary to add climate data.
[3] ANOVA and multiple comparison should be used to assess the scald reaction of these Turkish Barley varieties.
We conducted ANOVA on the field data.
[4] The full name of GGE should be “genotype plus genotype-by-environment” (L134-135).
Full name of GGE was corrected. (Line 138)
[5] The subscripted index should be in the equation (1), e.g., ɛij.
Subscripted index was corrected. (Line 151)
[6] The version of GGEBiplots should be added (L155).
Version of GGEBiplots package was added. (Line 158)
Reviewer 2 Report
Comments and Suggestions for Authors
Except for that the total content of the article is much less. This article was well composed and the experimental design was reasonable, the resistant lines would be useful in prduction and breeding program.
although the method and results of the article are acceptable, the content is really a little less, maybe it only could be published as a short communication
1. What is the main question addressed by the research?
The article aimed to identify varieties that resistant to a fungal pathogen which can cause scald during barley development.
2. Do you consider the topic original or relevant in the field? Does it address a specific gap in the field?
With the frequently changes of climate, the occurrence trend of the disease is more and more serious, while selection of resistant varieties is the best way to deal with the diseases. So the present study
3. What does it add to the subject area compared with other published material?
For the present ms , the method is the traditional identification of disease resistant, but the materials are different from others, and are local varieties.
4. What specific improvements should the authors consider regarding the methodology? What further controls should be considered?
In the beginning of results, the authors should explain the difference between repetitions, and make sure that the consistency of data between duplicates is relatively high.
5. Are the conclusions consistent with the evidence and arguments presented and do they address the main question posed?
yes
6. Are the references appropriate?
yes
7. Please include any additional comments on the tables and figures.
It is better to list the main information of 40 varieties in the materials and method part, thus the present table1 could be simplified
Author Response
responses below and the corresponding revisions/corrections in red changes in the revised file.
Comments and Suggestions for Authors
Except for that the total content of the article is much less. This article was well composed and the experimental design was reasonable, the resistant lines would be useful in prduction and breeding program. Although the method and results of the article are acceptable, the content is really a little less, maybe it only could be published as a short communication
- What is the main question addressed by the research?
This study aims to identify resistant varieties to a fungal pathogen which cause scald during barley development and to evaluate the reactions of the varieties to scald by using the new developed scale.
- Do you consider the topic original or relevant in the field? Does it address a specific gap in the field?
With the frequently changes of climate, the occurrence trend of the disease is more and more serious, while selection of resistant varieties is the best way to deal with the diseases. So the present study was conducted to determine the reactions of some Turkish barley varieties against scald, (ii) to identify the disease reaction groups such as immune, resistant, moderately resistant, moderately susceptible and susceptible, and (iii) to evaluate the identified reaction groups by performing GGE Biplot analysis.
- What does it add to the subject area compared with other published material?
For the present article, the method is different from the other disease rating scales, and also the materials are different from others, and are local varieties.
- What specific improvements should the authors consider regarding the methodology? What further controls should be considered?
In the beginning of results, the authors should explain the difference between repetitions, and make sure that the consistency of data between duplicates is relatively high.
We implemented an ANOVA which includes repetitions as a source of variance. Results of ANOVA can be used to evaluate the difference between repetitions, or whether the field data has been reliable. (Line 176-177).
- Are the conclusions consistent with the evidence and arguments presented and do they address the main question posed?
Yes
- Are the references appropriate?
Yes, and we also added new references in the manuscript. . (Line 326-332 and 347).
- Please include any additional comments on the tables and figures.
It is better to list the main information of 40 varieties in the materials and method part, thus the present table1 could be simplified
As suggested by you, we divided Table 1 into two tables. (Line 98 and 174).
Reviewer 3 Report
Comments and Suggestions for Authors
Dear colleagues! There are a few significant remarks that are recommended to be addressed before before the article is accepted for publication in the journal.
1. A key observation: did the authors conduct an assessment of barley grain yield? For breeding work it is very important to compare the degree of leaf lesions, the degree of resistance based on different evaluation scales and the final product - grain yield. Without data on yield as a function of disease severity, it is difficult to talk about varietal resistance. And if the authors have these data, they should be cited in the article. Especially since in section 1. Introduction it is stated that infection reduces yield and grain quality [4-6]. Losses can be up to 70% [7,8], the disease significantly reduces 1000 seed weight [10].
2. The literature review considers fundamental scientific works, but out of 37 works, only 5 sources are published in the last 5 years. In my opinion, it is necessary to expand the analysis of modern literature. Especially since there is a lot done in this area in the world. I recommend looking at the works:
1. Girma Ababa, Asela Kesho, Yitagesu Tadesse, Dereje Amare, Reviews of taxonomy, epidemiology, and management practices of the barley scald (Rhynchosporium graminicola) disease // Heliyon, Volume 9, Issue 3, 2023, e14315,https://doi.org/10.1016/j.heliyon.2023.e14315
2. M. Sakellariou, P.V. Mylona, New Uses for Traditional Crops : the Case of Barley Biofortification, 2020, https://doi.org/10.3390/agronomy10121964.
3. A. Backes, G. Guerriero, E. Ait Barka, C. Jacquard, Pyrenophora teres: taxonomy, morphology, interaction with barley, and mode of control, Front. Plant Sci. 12 (2021), 614951, https://doi.org/10.3389/fpls.2021.614951.
4. Abebe W. Barley net blotch disease management: A review //International Journal of Environmental & Agriculture Research. – 2021. – V. 7. – №. 9. – P. 69-81.
5. J.E. Cope, J. Norton, G.T.S. George, A.C. Newton, Identifying potential novel resistance to the foliar disease ‘Scald’(Rhynchosporium commune) in a population of Scottish Bere barley landrace (Hordeum vulgare L.), J. Plant Dis. Prot. 128 (2021) 999–1012, https://doi.org/10.1007/s41348-021-00470-x.
3. The authors studied 40 varieties at different ecological locations and in different years. Therefore, a brief characterisation of the weather, climate and soil conditions of the study regions should be given (can be brief). The article will be read by scientists from different countries - it will be difficult for them to understand the growing conditions in question.

Author Response
Thank you very much for taking the time to review this manuscript. Please find the detailed responses below and the corresponding revisions/corrections in red changes in the revised file.
Reviewer 3
Dear colleagues! There are a few significant remarks that are recommended to be addressed before the article is accepted for publication in the journal.
- A key observation: did the authors conduct an assessment of barley grain yield? For breeding work it is very important to compare the degree of leaf lesions, the degree of resistance based on different evaluation scales and the final product - grain yield. Without data on yield as a function of disease severity, it is difficult to talk about varietal resistance. And if the authors have these data, they should be cited in the article. Especially since in section 1. Introduction it is stated that infection reduces yield and grain quality [4-6]. Losses can be up to 70% [7,8], the disease significantly reduces 1000 seed weight [10].
The introduction section states the significance of the scald disease namely "it is stated that infection reduces yield and grain quality [4-6]. Losses can be up to 70% [7,8], the disease significantly reduces 1000 seed weight [10]" however, as indicated by the reviewer, the relationship between disease and grain yield was not studied in the current article. Therefore, we do not have data regarding this
- The literature review considers fundamental scientific works, but out of 37 works, only 5 sources are published in the last 5 years. In my opinion, it is necessary to expand the analysis of modern literature. Especially since there is a lot done in this area in the world. I recommend looking at the works:
As suggested by you, we added recently published references within the article. (Line 326-332 and 347).
Ababa, G.; Kesho, A.; Tadesse, Y.; Amare, D. Reviews of taxonomy, epidemiology, and management practices of the barley scald (Rhynchosporium graminicola) disease. Heliyon 2023.
Abebe, W. Barley net blotch disease management: A review. International Journal of Environmental & Agriculture Research 2021, 7, 69-81.
Backes, A.; Guerriero, G.; Ait Barka, E.; Jacquard, C. Pyrenophora teres: taxonomy, morphology, interaction with barley, and mode of control. Frontiers in plant science 2021, 12, 614951.
Cope, J.E.; Norton, G.J.; George, T.S.; Newton, A.C. Identifying potential novel resistance to the foliar disease ‘Scald’(Rhynchosporium commune) in a population of Scottish Bere barley landrace (Hordeum vulgare L.). Journal of Plant Diseases and Protection 2021, 128, 999-1012.
- The authors studied 40 varieties at different ecological locations and in different years. Therefore, a brief characterisation of the weather, climate and soil conditions of the study regions should be given (can be brief). The article will be read by scientists from different countries - it will be difficult for them to understand the growing conditions in question.
As the disease was artificially inoculated and its development was induced by moisture pressure in trials, the influence of climate can be disregarded. Therefore, we do not think that is necessary to add climate data.